# Bioinformatics Analysis of the Periodicity in Proteins with Coiled-Coil Structure—Enumerating All Decompositions of Sequence Periods

**DOI:** 10.3390/ijms23158692

**Published:** 2022-08-04

**Authors:** Andre Then, Haotian Zhang, Bashar Ibrahim, Stefan Schuster

**Affiliations:** 1Department of Bioinformatics, Matthias Schleiden Institute, University of Jena, Ernst-Abbe-Platz 2, 07743 Jena, Germany; 2Centre for Applied Mathematics and Bioinformatics, and Department of Mathematics and Natural Sciences, Gulf University for Science and Technology, Hawally 32093, Kuwait; 3European Virus Bioinformatics Center, Leutragraben 1, 07743 Jena, Germany

**Keywords:** coiled coil, decomposition algorithm, hydrophobic interaction, money-changing problem, recursive algorithm, hexaprenyl diphosphate synthase, seduheptulose-7-phosphate isomerase, aldehyde dehydrogenase

## Abstract

A coiled coil is a structural motif in proteins that consists of at least two α-helices wound around each other. For structural stabilization, these α-helices form interhelical contacts via their amino acid side chains. However, there are restrictions as to the distances along the amino acid sequence at which those contacts occur. As the spatial period of the α-helix is 3.6, the most frequent distances between hydrophobic contacts are 3, 4, and 7. Up to now, the multitude of possible decompositions of α-helices participating in coiled coils at these distances has not been explored systematically. Here, we present an algorithm that computes all non-redundant decompositions of sequence periods of hydrophobic amino acids into distances of 3, 4, and 7. Further, we examine which decompositions can be found in nature by analyzing the available data and taking a closer look at correlations between the properties of the coiled coil and its decomposition. We find that the availability of decompositions allowing for coiled-coil formation without putting too much strain on the α-helix geometry follows an oscillatory pattern in respect of period length. Our algorithm supplies the basis for exploring the possible decompositions of coiled coils of any period length.

## 1. Introduction

In 1953, Francis Crick described coiled coils as structures in fibrillary proteins consisting of two or three α-helices wound around each other to form a superhelix [1]. They are often classified as supersecondary structures. Pauling et al. [2] analyzed two protein structures and retrieved evidence for the pitch of the α-helical turn to be 5.15 Å instead of the expected 5.4 Å (1.5 Å per residue times 3.6 residues per helical turn). He explained the reduced height by the α-helices coiled together in a counter-clockwise orientation, opposite to the orientation of the single α-helices themselves (clockwise). Further, Crick predicted that apolar residues of the α-helices come into contact every 3–4 residues to stabilize the superhelix through hydrophobic “knobs-into-holes” interactions. This leads to a recurring sequence motif of seven residues called the heptad motif. This was experimentally confirmed by Hodges et al. [3] in 1973 through analyzing tropomyosin, an α-fibrous protein. In this sequence motif, the amino acids are repeatedly labeled from a to g, where a and d always comprise the amino acids that participate in interhelical interactions.

Since more and more protein crystal structures have become available, it has been shown that in reality the canonical heptad motif is not often as perfectly realized as Crick’s elegant theory suggested [4]. Irregularities such as “skips”, “stammers”, and “stutters” alter the canonical heptad motif, leading to structurally more demanding distances between interhelical interactions. Skips are insertions of one residue in the heptad motif and were first described in the end regions of nematode myosin [5]. Analogously three-residue-insertions, called stammers [6], and four-residue-insertions, called stutters [7], were discovered. Insertions can be localized within a single heptad, leading to decad- (7 + 3 = 10) and hendecad-motifs (7 + 4 = 11), or delocalized across multiple heptad repeats (for example 14 + 4 = 18). One of several methods to find non-canonical periods is by Fourier analysis [8]. Through partly destabilizing the superhelix structure, non-heptad motifs provide valuable contributions to protein functions. Examples include the oligosaccharide binding region in C-type mannose-binding protein containing a trimeric coiled coil [9] and induced dimerization upon coiled-coil stabilization through DNA-binding by GAL4 [10].

Coiled coils may also differ in their style of interhelical interaction. The two main modes are called “knobs-into-holes” interactions, where the hydrophobic residue of one α-helix is placed in-between four hydrophobic residues of the other helix, which form a hole, and “knobs-to-knobs” packing, where residues of neighbouring helices interact directly with each other. An example of the latter type is provided by the leucine zipper, in which two leucines of neighboring strands stand opposite to each other and form a hydrophobic interaction [11]. In the leucine zipper, both the period and the distance between contacts equals 7.

As coiled coils exhibit a high degree of geometric and functional diversity [12]–the basic principles of their interactions being also well understood in a quantitative sense [13], and public databases with a comparatively high number of crystallographic structures being available [14]–they are of particular interest in synthetic biology. Thus, a set of various toolkits for designing coiled coils with well-defined properties that are transferable between different contexts is desirable [15]. Discoveries of sequence–structure relationships resulted in rules guiding the process of protein engineering towards the desired properties [13,16]. Salt bridges often observed in the e-/g-region can for example be harnessed for designing orthogonal helix interaction by bringing together helices with complementary electrostatic charges [17], whereas hydrophobic amino acid propensities in the a-/d-region for hydrophobic interactions can guide the choice of amino acid for designing multimeric coiled coils of the desired degree [15].

Coiled coils have been constructed for applications such as controlled drug release, formation of nanotubes with desired structural properties, and immunological targeting [18]. In terms of developing guidelines for synthetic biology research, it is an interesting question to ask what regions of the potential property space of coiled coils have been realized by nature already and which of the remaining regions are to be prioritized in protein design.

As one key determinant of coiled-coil properties is the sequence of interhelical contact residues, we will present a method for calculating all possible decompositions for a given sequence period of interhelical contact amino acids, assuming that the allowed distances between interhelical contact residues are limited to 3, 4, and 7. We will outline relationships to the money-changing problem [19] and to decomposition methods in analyzing mass spectrometry data [20]. We further take a look at which periods have been realized in nature by searching and evaluating the registered coiled coils of the CC+-database [14]. Realized and theoretically possible–but so far undiscovered–coiled-coil periods are compared to discover hitherto unexploited potentials.

## 2. Results

From Sylvester’s Theorem, it follows that the largest number that cannot be composed from 3 and 4 is 3 × 4 – 3 − 4 = 5. Taking the number 7, in addition, does not change this result because it is larger than 5.

Moreover, there are some periods that can, in principle, be decomposed but drop out because all their decompositions are repetitions of shorter periods. This applies to 6 = 3 + 3, 8 = 4 + 4, 9 = 3 + 3 + 3 and 12 = 3 + 3 + 3 + 3 or 4 + 4 + 4.

The numbers of theoretically possible decompositions of interhelical contact residue distances as determined by our algorithm up to period 40 are shown in Figure 1. The numbers of decompositions observed in dimeric coiled coils listed in the CC+-database are shown there as well. The largest period we found in the database is 140 (PDB-ID: **2efs**). A complete list of all the theoretically possible dimeric coiled-coil decompositions is provided in Appendix A.

The results shown in Figure 1 agree with the above analytical results. Moreover, our numerical calculations support the conjecture that 12 is the largest period that cannot be decomposed in a non-redundant way. For example, 16 can be written as 4 × 4 but also as 4 × 3 + 4 or 3 × 3 + 7, so that it has two decompositions.

Overall, 88% of the dimeric coiled-coil decompositions in the CC+-db exclusively consist of interhelical contact residue distances we assumed for calculation of the theoretically possible decompositions–namely 3, 4, and 7. We want to emphasize that only residues that are identified by SOCKET as participating in knobs-into-holes packing are identified as contact residues. That means two amino acids with hydrophobic characteristics can be placed next to each other in the sequence without necessarily leading to a distance of one. The three most frequently occurring irregular interhelical contact residue distances are 1 (145 occurrences), 11 (122 occurrences), and 10 (91 occurrences). No decompositions were found in dimeric coiled coils with more than one repetition of distances 3 or 4 (for example 3 + 3 + 3). This also in part explains why there are no decompositions found in dimeric coiled coils for certain period lengths. Figure 2 shows the number of theoretical possible decompositions that do not contain a repeat of a hydrophobic contact distance of 3 or 4 longer than two. The oscillatory pattern is similar to the one observed for decompositions found in the CC+-db. For example, for period lengths 22 and 23, there are 9 and 12 possible decompositions, respectively (see Appendix A). However, of those only 3 and 2 do not contain such repeats. Therefore, the percentage of decompositions realized or realizable in nature is probably small for those period lengths.

Next, we evaluated the relationship between demanding decompositions (interhelical contact residue distances of 3 or 4 are repeated, 3 + 3 or 4 + 4) and structural properties of the coiled coils (see Figure 3). By comparing the means of demanding and non-demanding decompositions, we found that coiled coils of these two types do not differ significantly from each other in terms of local radius range (mean for demanding decompositions: 0.713 Å, mean for non-demanding decompositions: 0.622 Å, *p*-value of 0.139), but they do differ in terms of the highest local period deviation from 3.5 (mean values: 0.112 and 0.071, *p*-value of 0.012) and local period range (mean values: 0.109 and 0.085, *p*-value of 0.003). As it seems like demanding decompositions tend to cause both extreme local periodicity values and large fluctuations in local periodicity, we proceeded by having a closer look to determine whether the demanding regions of a decomposition coincide with local periodicity extremes.

In Figure 4, we depicted the ranges of local periodicity/radius for each coiled coil. It becomes obvious that most of the demanding decompositions consist entirely of short coiled coils with three interhelical contacts and either a 3 + 3 or a 4 + 4 repeat. For those decompositions it is naturally hardly possible to spot local disturbances caused by the 3 + 3 or 4 + 4 repeat. More interesting, however, are longer periods. In the following, we will examine the coiled coils of the proteins with PDB-IDs **3aqb**, **5i01**, and **6fjx**, as they show interesting patterns of a step-like rise in local periodicity, a valley of local periodicity with rising values at the edges of the coiled coil, and a continuous increase of local periodicity towards the 4 + 4 repeat, respectively.

**3aqb:** The PDB-ID represents the heterodimeric hexaprenyl diphosphate synthase from Micrococcus luteus [21]. The protein consists mainly of α-helix and coiled-coil regions. The coiled coil with a decad motif and the decomposition [4, 3, 3] is formed between α-helices of the large and small subunits of the enzyme. The small subunit was shown to regulate the chain length of the product [21]. Figure 5 shows the coiled coil being part of the ligand-binding pocket. Towards the end of the coiled coil facing the ligand, the distance between the α-helices increases. Different distance cut-offs might explain why this region is still assigned to the coiled coil by the CC+-database, although it is not part of the structural analysis in the CCdb.

**5i01:** The protein GmhA is the seduheptulose-7-phosphate isomerase of the pathogenic bacterium *Neisseria gonorrhoeae* [22]. It is a potential target for antibiotic treatment and consists mainly of α-helices and coiled coils, but also has some β-sheets. The coiled coil is located at the interface between two of the four monomers at the protein surface (see Figure 6). Interestingly, at the end of the demanding 4 + 4-region the α-helix of the coiled coil exhibits a kink and continues at an angle of around 60°. This kink is probably also the reason why CCdb does not cover the last two residues participating in the coiled coil, as the transition zone is a borderline case between an orderly coiled coil and a disordered region.

**6fjx:** The PDB-ID contains the aldehyde dehydrogenase of *Thermus thermophilus* [23]. Upon reduction of NAD(P) the enzyme oxidizes a broad range of aldehydes into the corresponding carboxylic acids. The coiled coil is located at the surface of the protein (see Figure 7). As in PDB-ID 5i01, the [4, 4]-repeat is located at the kink of the coiled coil. The kink in combination with the high pitch angle, i.e., the high rotational displacement of the α-helices from a parallel alignment, causes CCdb to only recognize a small window of the entire coiled coil.

## 3. Discussion

We developed a novel algorithm for enlisting all possible decompositions consisting of the most prevalent distances between interhelical contacts, namely 3, 4, and 7, for any coiled-coil period length. We considered hydrophobic and knobs-into-hole contacts. In the case of trimeric or multimeric coiled coils, we considered the contacts of one given helix to one given neighboring helix. Interactions to other helices in the coiled coil can, subsequently, be analyzed in an analogous way. It is worth noting that Fourier analysis, while often used in analyzing periods in coiled coils [8] is of limited use for our study because it only provides average distances rather than a sequence of different distances.

The most basic problem comparable to our task is the money-changing problem [19] and an even more related problem is the exhaustive enumeration of all weighted strings indicating the possible fragmentation of a mass *M* in mass spectrometry [20]. In writing our algorithm, we have aimed at simplicity rather than high efficiency. Running time is not a serious problem here because the sequence periods are limited. The largest period we found is 140 (PDB-ID: **2efs**).

Periods enabling left-handed coiled coils (e.g., 7, 10, 14) can be observed particularly often in proteins that have to withstand tensile stress, such as α-keratin, fibrin and cohesin. The opposite handedness of the single helix and coiled coil is also characteristic for man-made ropes since it makes them more resistant to unwinding upon tension. In contrast, coiled coils in many other proteins, which fulfill other functions such as surface adhesion or enzyme catalysis, are right-handed or not markedly wound. For example, the coiled coils in hemagglutinin of the influenza virus [4] or the seduheptulose-7-phosphate isomerase analyzed above do not have any handedness and, thus, could rather be considered as helix bundles.

The fraction of decompositions observed in coiled-coil dimers with respect to the theoretically possible decompositions declines with increasing period length. This is partly due to the declining proportion of non-demanding decompositions, i.e., decompositions which do not contain two or more insertions of stammers (three residues inserted into the heptad pattern) or stutters (four residues inserted into the heptad pattern) next to each other. The corresponding oscillatory patterns (see Figure 2) we analyzed in the Results section support this hypothesis. Observing the period lengths of the identified demanding decompositions gives another explanation. No demanding decompositions for a period length over 15 have been identified. The examples we covered show that especially 4 + 4-repeats often lead to a kink in the corresponding α-helix. As the continuation of the coiled coil requires the partner α-helix to have a similar kink, the probability to observe decompositions with 4 + 4- and/or 3 + 3-repeats in the middle of the decomposition is lowered.

As mentioned in the Introduction, amino acids in coiled coils are often repeatedly labeled from a to g. This has the drawback that it could be confused with the one-letter code for amino acids. Moreover, it is more difficult to count them as if they were labeled by 0, 1, 2, etc. We advocate using such numeric labeling (although it is of minor importance in the present paper). Starting with 0 rather than 1 has the advantage that the next amino acid, second-next, etc., can easily be determined.

Our complete enumeration of all possible decompositions for a given period length gives a hint as to which period lengths have the most existing design options in synthetic biology. Demanding decompositions, especially, can extend the versatility in structure and function of coiled-coil designs [24]. A kink like the ones we observed in **5i01** and **6fjx** has been shown to be important for DNA binding during mismatch repair [25]. Stammer regions in demanding decompositions are also able to convey specificity of binding by requiring the same phase of stammer insertion in the binding partner [26]. Despite these static functions, the local un- or overwinding in demanding decompositions caused by [4, 4]- and [3, 3]-repeats, respectively, allows for interesting properties in terms of the dynamic behavior of coiled coils. For example, it may affect the accessibility of the helix core for ligand binding and thereby have an impact on enzymatic reaction rates [27] or allow for further non-covalent interactions between the helices [28]. In addition, metastability caused by a stammer in the coiled coil of L1ORF1p, an RNA-binding protein, has been shown to be crucial for retrotransposition of human LINE-1 [29]. However, this is a hypothesis based on limited data, and further research is required to elucidate the connection between demanding decompositions and the structural properties of the corresponding coiled coils.

## 4. Methods

### 4.1. Defining the Problem

Our aim is to establish a method for calculating all linear combinations of the numbers 3, 4, and 7 to obtain given periods of interhelical contacts. In these combinations, the sequence matters up to rotational shifts. For example, 3 + 4 is equivalent to 4 + 3, while 3 + 3 + 4 + 4 is not equivalent to 3 + 4 + 3 + 4. The latter would be equivalent, however, to 3 + 4 because 7 is the smallest period in this case.

The distances are here defined as the number of amino acids along a protein chain until the next amino acid forming a hydrophobic contact or, in the other interaction mode, a knob. For example, if a residue at position *i* is a knob residue and the next knob residue along the chain is at position *i* + 3, the distance between interhelical contact residues is 3. Some coiled coils include pairs of neighboring residues forming contacts to another helix. That is, the distance equals one. A possible decomposition would then be, for example, 3 + 1 + 3. However, taking 1 as possible distance, far too many unrealistic decompositions would be obtained, such as 3 + 1 + 1 + … + 1. Therefore, in the case of such pairs, we only consider, from any pair, the residue that is closest to the N-terminus of the protein. 3 + 1 + 3, for example, would then be written as 4 + 3.

It is worth mentioning that whether a feature is hydrophobic is not defined in a clear-cut way. As it depends on the dipole moment of the side chain, different classifications are found in the literature according to the threshold value of the dipole moment below which the amino acid is considered to be hydrophobic [30]. Amino acids that are considered hydrophobic or hydrophilic depending on the classification scheme include histidine, cysteine and tyrosine. While threonine is usually considered hydrophilic due to its hydroxy group, it does participate in knob-into-holes interactions due to its methyl group [4]. In our approach, we assume that a set of amino acids forming hydrophobic interactions or acting as knobs is given at the outset, regardless of how it is defined.

### 4.2. Mathematical Background

A mathematical problem comparable to the task treated here is the money-changing problem [19,31]. In this problem, for a given positive integer amount *W* (for example, money), all combinations of elements (e.g., coins) from a set consisting only of positive integer values (*x*_1_, *x*_2_, …, *x_n_*) are searched for, which add up to *W*. The elements may occur more than once in the sum. In a mathematically concise definition, they must fulfill the equation $\overset{n}{\underset{j=1}{\sum }}a_{j}x_{j}=W$. *a_j_* is the non-negative integer weighting factor, indicating how often the element *x_j_* is part of the combination. In mathematics, equations with integer variables are called Diophantine equations.

It is obvious that, as a tendency, the larger *W* is, the more possible decompositions of *W* exist. The largest number that cannot be decomposed for a given set of composing numbers is called the Frobenius number after mathematician Ferdinand Frobenius [32]. One of several algorithms for solving the money-changing problem was presented by Böcker and Lipták (2007) [31].

For the case *n* = 2, James Joseph Sylvester derived the following result [33]:

Sylvester’s Theorem: Let *x*_1_ and *x*_2_ be two natural numbers that are prime to each other (that is, they do not have any common divisor greater than one). Then, the largest natural number *n* that cannot be written in the form *n* = *a*_1_
*x*_1_ + *a*_2_
*x*_2_ with non-negative integer numbers *a*_1_ and *a*_2_, is *x*_1_
*x*_2_ − *x*_1_ − *x*_2_.

For the case *n* > 2, no general results have been derived so far. A rather trivial result for the case *n* = 3 is that if *x*_3_ is larger than *x*_1_
*x*_2_ − *x*_1_ − *x*_2_, the Frobenius number is still given by the latter expression.

There is an important difference to the problem in question here because, in the money-changing problem, the order of the terms in the sum does not matter, while it does in the decomposition of sequence periods. Related problems are tackled in the analysis of mass spectrometry data [20]. It is of interest to decompose a given mass *M* into smaller masses of certain fragments. For certain applications (e.g., in metabolomics), this is done without considering the sequence of fragments, so that methods for solving the money-changing problem can be used [31]. For other applications (e.g., in analyzing peptides), the sequence does matter. In those cases, the concept of weighted strings can be used. This is nearer to our problem.

An alphabet of *k* characters, *σ_i_* (e.g., for amino acids) and a set of associated masses, *a_i_*, can be defined [20]. Each string then has its own total mass. Now we search for all strings whose masses sum up to *M*. Counting (but not yet enumerating) those strings can be performed by the following recursion. The number, *x*[*m*], of strings with mass *m* is given by $x_{i}=\overset{k}{\underset{i=1}{\sum }}x\left[m-a_{i}\right]$ with the initial condition *x*_0_ = 1. The idea is that each such string can be divided into a string that is one character shorter, plus one character (with mass *a_i_*). The open-access e-book by Böcker [20] (in its current state) only presents an algorithm for enumeration in the case without consideration of the sequence of fragments (so-called compomers) rather than for weighted strings [20].

### 4.3. Calculation of Possible Decompositions for Each Coiled-Coil Period Length

The calculation of the theoretically possible decompositions for each period length was coded in Python 3.6. No additional packages were required. The pseudocode, which can be easily implemented in any programming language, is shown in Algorithm 1.

The algorithm consists of a for-loop starting with the minimum period length *i* = 3 up to the user-defined maximum period length. First the length under consideration in the loop is compared with the set of allowed amino acid distances for interhelical contacts. If it is an element of the set, that single distance is a valid decomposition. If *i* is larger than, or equal to the sum of the two smallest elements in that set, the concatenation (conjunction) of all (previously calculated) decompositions of the period lengths *x* and *i* − *x*, with 3 ≤ *x* ≤ *i*/2, are examined in a for-loop with loop variable *x*. Those concatenations are accepted as valid decompositions for period length *i*, if they cannot be described as a repetition of decompositions of smaller period lengths and are not already contained in the concatenations computed previously in that step of the for-loop with variable *i*. Note that, for example, [3, 3, 4] can be obtained as a concatenation of [3, 3] and [4], but is obtained earlier by [3] and [3, 4]. Upon accepting a solution for *i*, all possible ways of writing the solution (rotational copies) are stored while only one of them is finally given in the output, since we consider sequences with repeating motifs. This means that for a period length of 11, for example, only one of the three decompositions [3, 3, 4], [3, 4, 3], and [4, 3, 3] is kept in the final set of solutions. That all rotational copies are stored in the course of the algorithm has the advantage that decompositions computed later can be compared with them.
**Algorithm 1:** Pseudocode of a program for calculating all theoretically possible decompositions (weighted strings) of given periods into 3, 4, and 7
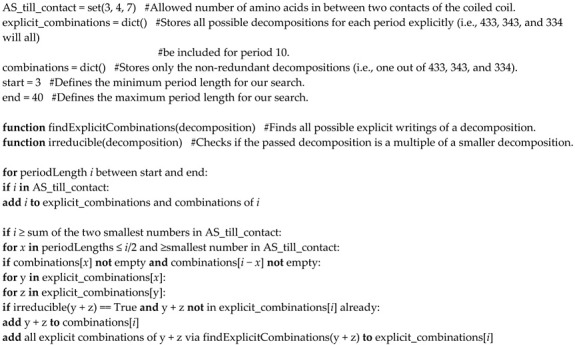


### 4.4. Data Analysis for Comparison with Naturally Occurring Coiled Coils

We extracted the information on dimeric coiled coils in the CC+-database [14] using python 3.6 and the mechanicalsoup package for webcrawling. We first used the dynamic search interface of the website (Figure 8) to retrieve a txt-file specifying the PDB-, CC-, and Helix-IDs as well as the amino acid positions at the start and end of the participating α-helices. Subsequently we used these entries as an input to crawl the website to download detailed information on each coiled coil stored in the SOCKET-format [34].

SOCKET identifies knobs-into-holes packing by checking if an amino acid residue of one helix is located in a hole formed by four amino acids of the partner helix. A hole is formed if the center of mass of the side chains of the four-hole residues are located in a distance to the center of mass of the knob residue not exceeding a specified distance cutoff. Although coiled-coil interactions are often referred to in a generalization as hydrophobic interactions [35,36], in principle any amino acid can participate in knobs-into-holes interactions [37]. Polar amino acids can often participate in interhelical interactions by their polar parts protruding over the edge of a hole. These interactions are categorized in SOCKET as Type 1 and 3 interactions and are sometimes also called knobs-across-holes interactions [37]. A further distinction between coiled-coil interactions in SOCKET is between complementary (Type 1 and 2) and non-complementary (Type 3 and 4) knob residues. A knob residue is called complementary if it is also a hole residue for a knob residue of an interacting helix. To identify as many non-canonical coiled coils as possible, we included any type of knobs into our analysis that SOCKET identifies by the default center of mass distance cutoff of 7.0 Å. The distances along the chains between those knob residues give the decomposition for the given period length. In the case of heteromeric coiled coils with different decompositions for each chain, we consider them both separately in our analysis.

### 4.5. Connecting the Decompositions to the Local Structural Properties of Coiled Coils

To evaluate how the different decompositions affect the structure of the coiled coil, we employed the CCdb, a non-redundant database of coiled-coil structures analyzed with SamCC-Turbo [38]. Via geometrical analysis, SamCC-Turbo can quantify local structural properties along the coiled coil. As we were particularly interested in gaining an understanding of how demanding the different decompositions are for building a coiled-coil structure, we focused on local periodicity and radius as two important parameters. We identified the right coiled-coil bundle for each structure by searching in the dataset for the corresponding PDB-ID and aligning start and end positions with a maximal tolerated deviation of four residues for any of the two start and two end positions. To consider this deviation is important, as pinpointing the exact position where the coiled coil starts/ends is to some extent ambiguous. Of the 3134 dimeric coiled coils identified with CC+, we were able to retrieve the structural information in CCdb for 2284. These remaining entries in our dataset are confirmed by both databases and are therefore assured to be valid coiled-coil structures.

### 4.6. Statistics

For assessing whether there is a difference between the structural properties of coiled coils with demanding decompositions, i.e., decompositions with at least one 3 + 3 or 4 + 4 repeat in the distance of knob residues in knobs-into-holes packing, and those with non-demanding decompositions, we employed the ttest_ind function of scipy 1.6.2. This function allows the use of the standard *t*-test for populations with equal variances or Welch’s *t*-test for populations, which does not assume equal variances. Accordingly, before the *t*-test was performed, we made use of the levene function of scipy to test for equal variances of the populations and chose the *t*-test accordingly.

### 4.7. Figures

Figure 1 and Figure 2 were made with Microsoft Excel 2019. Figure 3 was produced in python’s matplotlib library, version 3.3.4.

## 5. Conclusions

We report a novel algorithm for enumerating all decompositions of sequence periods in coiled-coil proteins. Examples of exotic decompositions conveying interesting functions have been discussed. Those functions are often linked to the dynamic behavior of coiled coils. Our method provides the basis for a systematic analysis of sequence period decompositions. A connection between the oscillatory pattern of the number of realized decompositions and the fraction of theoretically possible decompositions that do not contain two or more stutters or stammers next to each other has been observed. The design of demanding decompositions is especially deemed to be of interest for further development in protein design, as their metastability allows for interesting functions that often cannot be provided by regular heptad pattern decompositions. Furthermore, an extension of the theoretical analysis towards coiled coils with three or more participating α-helices might be interesting to determine whether the increased number of non-covalent interactions in those structures allows for more complex decompositions to be realized.

## Figures and Tables

**Figure 1 ijms-23-08692-f001:**
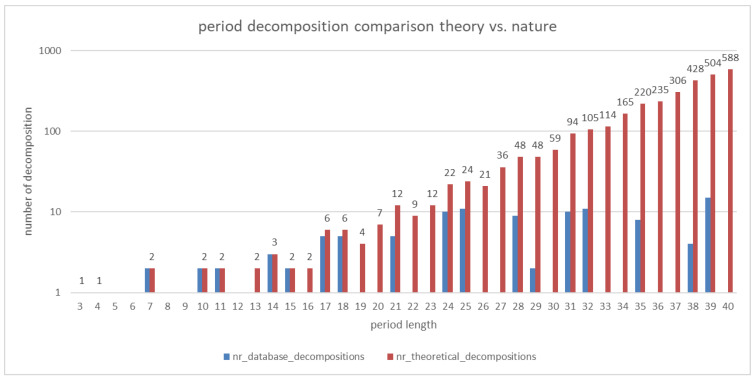
Comparison of the number of theoretically possible decompositions (red) and decompositions found in dimeric coiled coils listed in the CC+-database (blue) for period lengths between 3 to 40. Be aware that the y-axis (number of decompositions) is log-transformed.

**Figure 2 ijms-23-08692-f002:**
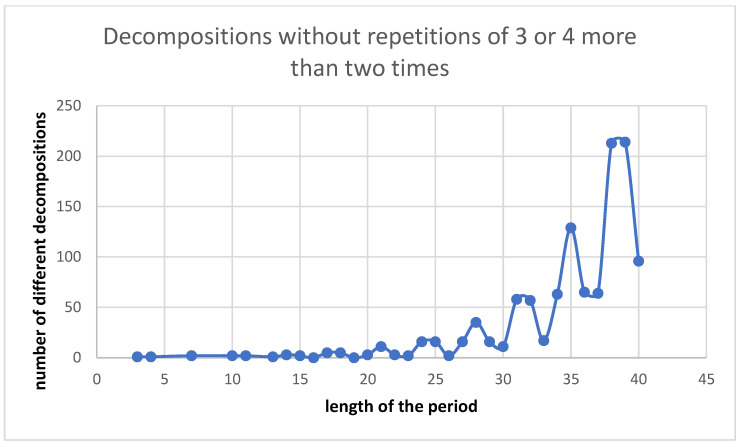
Decompositions not containing repeats of interhelical contact residue distances of 3 or 4 longer than two.

**Figure 3 ijms-23-08692-f003:**
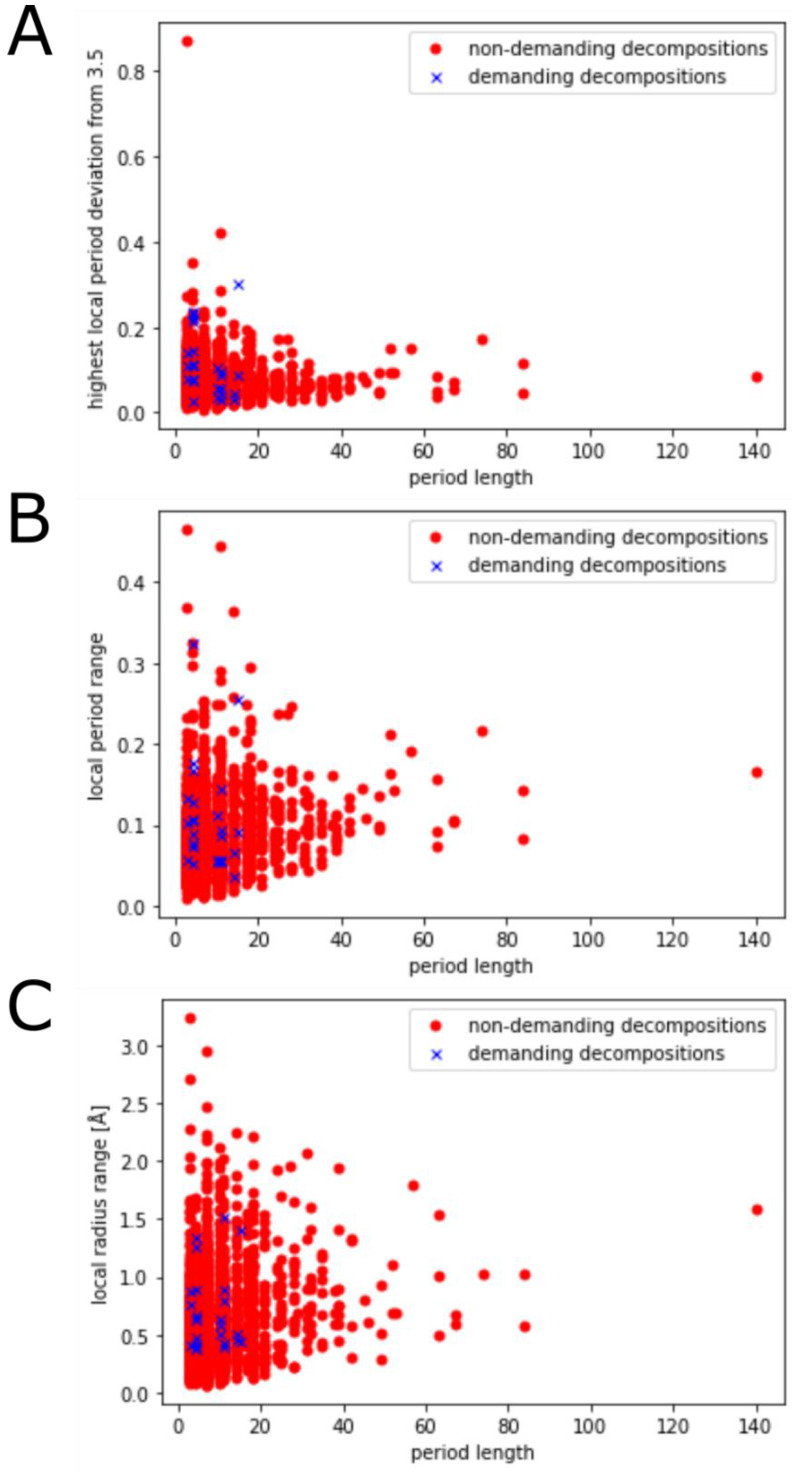
Comparison of structural properties between coiled coils with demanding (at least one repeat of interhelical contact distance 3 or 4) and non-demanding decompositions. (**A**): Comparing the highest local period deviation, i.e., the highest deviation along the coiled coil from the base periodicity of 3.5 Å. (**B**): Comparing the difference between the maximum local periodicity and the minimum periodicity along the coiled-coil structure. (**C**): Comparing the difference between the maximum local radius and the minimum local radius along the coiled-coil structure.

**Figure 4 ijms-23-08692-f004:**
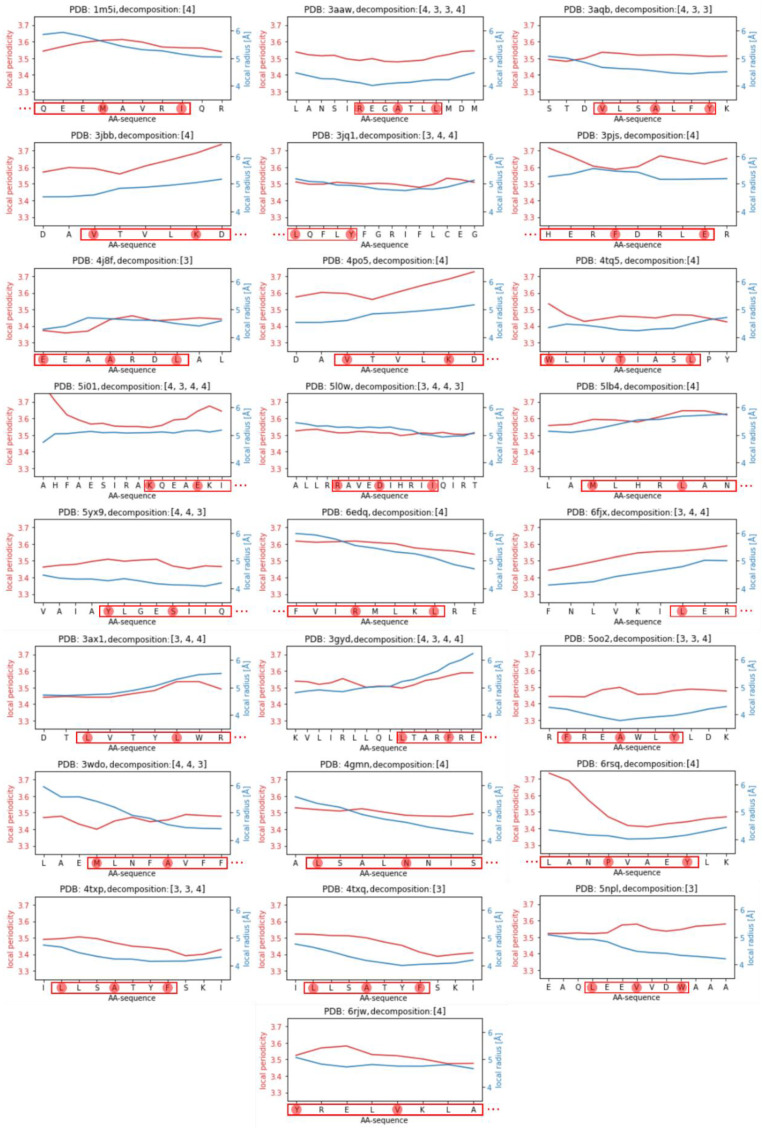
Local periodicity and local radius for the 25 identified demanding decompositions. The abscissa shows the amino acid sequence. The amino acids that are part of the 3 + 3 or 4 + 4 hydrophobic distance repeats within the decompositions are surrounded by a red box. Amino acids participating in interhelical contacts are further underlaid by a red circle. The three dots at the end of an amino acid sequence range indicate that the full coiled-coil sequence according to the CC+-database is deprecated in the CCdb.

**Figure 5 ijms-23-08692-f005:**
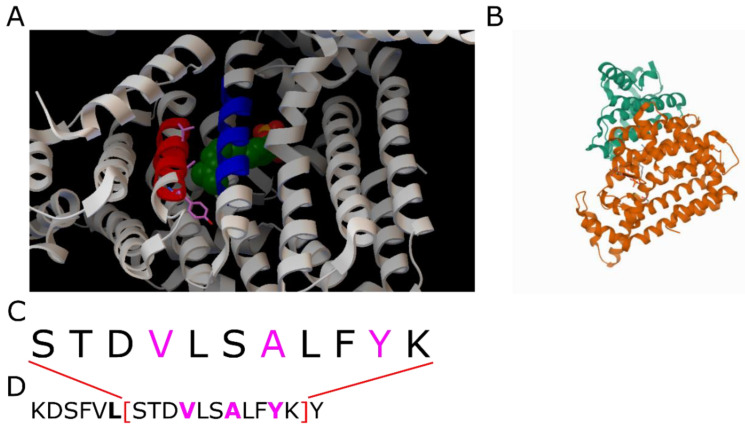
PDB-ID 3aqc, the ligand-bound form of 3aqb. (**A**): Detailed view on the coiled coil. The red α-helix realizes the demanding [4, 3, 3]-decomposition. The blue α-helix is the interaction partner for the formation of the coiled coil. (**B**) Shows a total view on the protein with different colors for its subunits. (**C**): Amino acid sequence of the coiled-coil range analyzed by the CCdb. (**D**): Complete sequence of the α-helix. Contact residues of the coiled coil are bold. The contact residues of the [3, 3]-repeat are colored in magenta, corresponding to (**A**).

**Figure 6 ijms-23-08692-f006:**
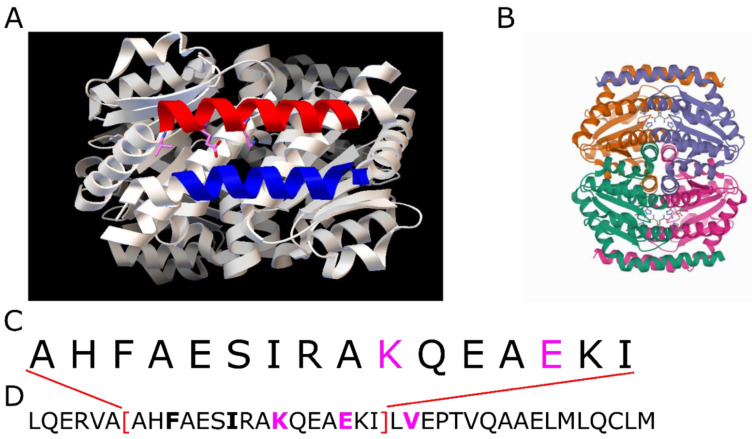
PDB-ID 5i01. (**A**): Detailed view on the coiled coil. The red α-helix exhibits the demanding [4, 3, 4, 4]-decomposition. The blue α-helix is the interaction partner for the formation of the coiled coil. (**B**) Shows a total view on the protein with different colors for its subunits. (**C**): Amino acid sequence of the coiled-coil range analyzed by the CCdb. (**D**): Complete sequence of the α-helix. Contact residues of the coiled coil are bold. The contact residues of the [4, 4]-repeat are colored in magenta, corresponding to (**A**).

**Figure 7 ijms-23-08692-f007:**
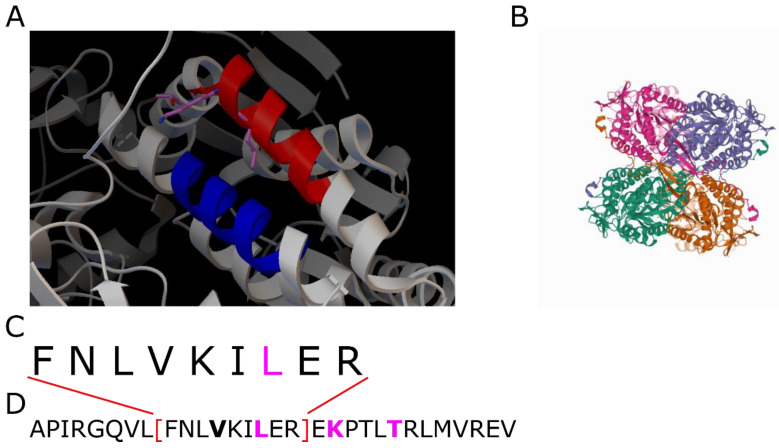
PDB-ID 6fjx. (**A**): Detailed view on the coiled coil. The red α-helix exhibits the demanding [3, 4, 4]-decomposition. The blue α-helix is the interaction partner for the formation of the coiled coil. (**B**) Shows a total view on the protein with different colors for its subunits. (**C**): Amino acid sequence of the coiled-coil range analyzed by the CCdb. (**D**): Complete sequence of the α-helix. Contact residues of the coiled coil are bold. The contact residues of the [4, 4]-repeat are colored in magenta, corresponding to (**A**).

**Figure 8 ijms-23-08692-f008:**
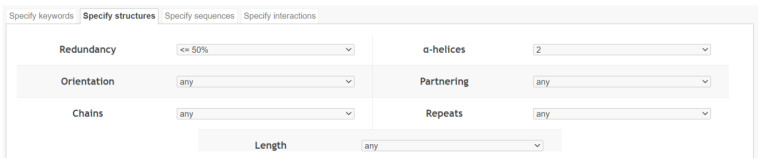
Input parameters for the CC+ dynamic search interface. Retrieval of any non-redundant dimeric coiled coils.

## Data Availability

The authors confirm that the data supporting the findings of this study are available within the article and its Appendix A.

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
