# Peer review of "Bioinformatics Analysis of the Periodicity in Proteins with Coiled-Coil Structure—Enumerating All Decompositions of Sequence Periods"

_ijms, 2022, doi:10.3390/ijms23158692_

Round 1
Reviewer 1 Report
The presented manuscript descibes the periodicity of coiled-coil structure in proteins, which was studied by bioinformatic tools. The obtained results were, in my opinion , clearly described and presented. I have no objections to the overal merit, and the obtained data have been described in a logical sequence. Therefore, I recommend this manuscript for publication after minor corrections e.g. the formatting of references (some references have a doi number, and some do not)
Author Response
We thank the reviewer very much for her/his careful reading of our manuscript. We checked and corrected the doi in the references.
Reviewer 2 Report
The manuscript "Bioinformatics analysis of the periodicity in proteins with coiled-coil structure - Enumerating all decompositions of sequence periods" by Andre Then et al. presents the development and application of an algorithm for enumerating all decompositions of sequence periods in coiled-coil proteins. Owing to their high structural and functional diversity, coiled coils have been employed in numerous applications of synthetic biology (controlled drug release, designed nanotubes, applications in immunology). Thus, this work aims to reveal the sub-space of all possible of coiled-coil properties that has been realized already, and more interestingly, which of the remaining is the most interesting for protein design.
The authors wrote a very nice introduction (background) and present their methodology in such a clear way that it can be replicated even by a non-expert in the field. The mathematical background in particular is presented in a very comprehensive manner. Statistics are applied appropriately and high quality plots and figures illustrate the results nicely. Analysis of the resulting decompositions has been carried out carefully in comparison with dimeric coiled coils retrieved from the CC+-database so as to uncover the fraction of theoretically possible decompositions realized.
In addition, the authors found that demanding decompositions display higher local period deviation from 3.5 and higher difference from local period range with respect to non-demanding decompositions, a finding that can be attractive in protein design. On these grounds, the authors have examined the structural implications of coiled-coils in 3 enzymes that exhibited interesting patterns in local periodicity. This analysis provide hints about structural and dynamical implications of demanding decompositions caused by [4, 4] and [3, 3] repeats, which might play key roles in enzymatic activity and ligand binding.
Overall this work has been carried out, analyzed and presented in an exemplary way. I reckon that the findings of this study will be of interest to the readers of IJMS who work in the fields of Bioinformatics, Biochemistry and Protein Design. Therefore I find merit and I suggest publication of this work at its present form.
Author Response
We thank the Reviewer for their positive comment and careful review,